# 3D Bioprinting Using Hydrogels: Cell Inks and Tissue Engineering Applications

**DOI:** 10.3390/pharmaceutics14122596

**Published:** 2022-11-24

**Authors:** Annika C. Dell, Grayson Wagner, Jason Own, John P. Geibel

**Affiliations:** 1The John B. Pierce Laboratory, Inc., New Haven, CT 06519, USA; 2Fraunhofer IMTE, Fraunhofer Research Institution for Individualized and Cell-Based Medical Engineering, 23562 Lübeck, Germany; 3Yale University, New Haven, CT 06520, USA; 4Yale University School of Medicine, New Haven, CT 06510, USA

**Keywords:** bioprinting, hydrogel, inkjet bioprinting, extrusion bioprinting, tissue engineering

## Abstract

3D bioprinting is transforming tissue engineering in medicine by providing novel methods that are precise and highly customizable to create biological tissues. The selection of a “cell ink”, a printable formulation, is an integral part of adapting 3D bioprinting processes to allow for process optimization and customization related to the target tissue. Bioprinting hydrogels allows for tailorable material, physical, chemical, and biological properties of the cell ink and is suited for biomedical applications. Hydrogel-based cell ink formulations are a promising option for the variety of techniques with which bioprinting can be achieved. In this review, we will examine some of the current hydrogel-based cell inks used in bioprinting, as well as their use in current and proposed future bioprinting methods. We will highlight some of the biological applications and discuss the development of new hydrogels and methods that can incorporate the completed print into the tissue or organ of interest.

## 1. Introduction

The strategies of engineering tissues and creating functional biological constructs that can replace or repair damaged tissues have advanced steadily over the last years and have impacted a wide spectrum of medical specialties, including the bioprinting of vascular conduits [1], bone implants [2], skin grafts [3], intestinal grafts [4], and cardiac tissues [5]. Though some current interventional methods in these fields have been successful, tissue engineering continues to offer new perspectives and alternatives to combat the shortcomings of these conventional surgical methods. In vascular and endovascular surgery, common interventions include stent implantation via endovascular interventions or bypass surgeries. Some of the downsides of these operations include vascular graft infections [6,7], in-stent restenosis [8,9], inflammation [9], stent fracture [10], and stent migration [11,12]. Additional complexities of these interventions include the presence of a foreign body, such as synthetic grafts or stents, for the patient’s lifetime. In addition, tissue engineering in vascular surgery offers a solution to the continued shortage of transplantable organs and tissues [13].

This review briefly presents polymeric hydrogel-based scaffolds used in tissue engineering and focuses on hydrogel-based cell ink formulations used in bioprinting, the characteristics of hydrogels with respect to their use in cell inks, the various bioprinting strategies, as well as some applications of hydrogel-based bioprinted structures. We present a general overview of the current hydrogel-based bioprinting applications and characteristics of the hydrogels used. Additionally, this work tends towards applications in vascular and endovascular surgery, as this is the area of expertise of this working group. Vascular structures play an important part in tissue engineering, as tissue vascularization is known to be a prerequisite for the growth of larger engineered tissues in vivo, including organs [14,15]. Additionally, vasculature is a structure that is often befallen with numerous pathologies and is conveniently simple in geometry and cell structure, providing an ideal proof of concept and an indicator of feasibility for further creation of more complex bio-printed structures [15,16].

The main technologies used for deposition and patterning of biological materials are inkjet (jet-based, drop-on-demand), microextrusion, and laser-assisted printing [17]. This review centers around the use of hydrogel-based cell inks in extrusion bioprinting and jet-based bioprinting, though other bioprinting modalities exist as well. Based on our experience, the two aforementioned bioprinting modes are the most promising types of bioprinting with regard to bioprinting anatomical and physiological structures for implantation purposes, which is why we focus on these. We refer to a number of other excellent reviews for further information on other types of bioprinting [17,18,19].

## 2. Properties of Polymeric Hydrogels

Polymeric hydrogels are three-dimensional crosslinked networks of hydrophilic polymer chains that are capable of holding substantial amounts of water, swelling up to 99% water (*w*/*w*) of their dry weight without dissolution [20,21]. Though other types of hydrogels and various methods of gelation exist, this work focuses on polymeric hydrogels. Polymeric hydrogels are desirable for 3D bioprinting and various tissue engineering applications due to their biocompatibility and tissue-like mechanical properties [1]. Hydrogels effectively reproduce the extracellular matrix, the natural environment of cells, and provide a hydrated and structurally supportive environment that can be efficiently and homogenously seeded with cells [21]. Cells can be distributed within this polymeric hydrogel to create a cell ink and extruded in a controlled fashion during bioprinting applications. Recently, our group and others have mixed cells into a hydrogel solution and extruded this cell ink to create cell-laden vascular structures [1].

Hydrogels are often composed of shear-thinning materials that have the ability to extrude under high shear stress while maintaining their mechanical properties afterwards [21,22]. This makes them desirable for bioprinting applications. Materials such as gelatin, polyethylene glycol (PEG), and Pluronic^®^ will behave as liquids during printing and then revert back to a gel-like structure once extruded, which provides the print with the necessary support to remain in the desired structure. A summary of the types of polymer hydrogels discussed in this work can be found in Table 1. The following subsections describe properties of natural and synthetic polymer hydrogels commonly used in bioprinting applications.

### 2.1. Physiochemical Properties of Hydrogels

An understanding of the physicochemical properties of hydrogels is crucial to gauge the stability, functionality, and toxicity of the applications of hydrogels in bioprinting. Key physicochemical properties highlighted in this review include pH, temperature, and cross-linking.

#### 2.1.1. pH

Most hydrogels are stored and printed at around the physiological pH (~7.4) [47,48]. The pH of hydrogels largely influences the swelling capacity of the hydrogels [48]. Swelling capacity dictates a hydrogel’s shape and volume changes; therefore, higher swelling capacity is preferred due to increased stability of the hydrogel [30]. Most hydrogels display the highest swelling capacity at around pH~7.4, when compared to both an acidic and basic medium [48].

A shift in pH usually results in a change in the charge of the polymer chains which leads to either swelling or deswelling of the hydrogel and an overall change in stability [49]. In particular, pH sensitive hydrogels are susceptible to changes in pH largely due to their ionic nature [49]. Cationic hydrogels tend to swell at low pH due to protonation of amino/imine groups, while anionic hydrogels swell at higher pH values due to ionization of acidic groups [49].

#### 2.1.2. Temperature

Temperature correlates with hydrogel viscosity inversely [50]. The higher temperature of the environment, the lower the viscosity, which correlates to less shear stress and less damage to the cells [50]. However, in the case of bioprinting, printing temperature depends on the type of polymer that is being used. For reactive printing of ionotropic polymers, the polymer solution for hydrogel formation can be stored and printed at cell culture temperature (37 °C) [51]. Because reactive printing of ionotropic polymers involves inducing gelation in a bath containing appropriate counter-ions, gelation is very fast and it is possible to print polymer solutions with cell culture media [51].

In terms of hydrogel that form from polymers that interact through physical associations, the optimal temperature varies depending on the type of polymer that is being gelated. Typically, hot solutions of these polymers are printed into a cooled environment where they then reach their gel transition temperature and undergo solidification [51]. For example, agarose is water-soluble at temperatures above 65 °C and has a gel melting temperature at 85 °C [52]. Therefore, agarose is typically stored in the printer reservoir at temperatures between 60 and 80 °C [51]. The agarose is then printed into a cool bath that can range from 17–40 °C, which lies below its gel transition temperature [19]. In polymers that gelate through physical associations, the final temperature of the gel often prevents it from being able to incorporate cells both during and immediately after printing since the temperature is not within the normal body temperature range and could be harmful to the cells.

#### 2.1.3. Crosslinking

Cross-linking is a post-printing procedure that modifies the internal structure of the printed hydrogel to maintain the structural integrity and achieve the desired mechanical properties of the bioprinted construct [50]. The two most common cross-linking mechanisms are physical cross-linking and chemical cross-linking [53]. Physical cross-linking is carried out by physical procedures and involves intermolecular interactions between polymer chains such as hydrophobic interactions, electrostatic interactions, hydrogen bonding, stereocomplexation, and guest-host interactions [53]. Physical cross-linking is reversible and little to no chemical reaction is involved in the preparation of this linking. Chemical cross-linking involves the addition of reacting agents to induce covalent bonding of chemically reactive functional groups [53]. Chemical cross-linking of hydrogels is irreversible, but its advantages are stability, tunable structures, and sound mechanical properties [53]. A less popular cross-linking mechanism is thermal cross-linking, and it relies on temperature changes in the surrounding environment. Most natural polymer hydrogels perform cross-linking at 37 °C [53]. However, there are few hydrogels (e.g., alginate, gelatin) that cross-link at room temperature [53].

### 2.2. Biological Properties of Hydrogels as Cell Inks

Hydrogels are useful in bioprinting because they share many similar features with natural extracellular membrane components and allow cell encapsulation in both a highly hydrated and mechanically supportive 3D environment [54]. The hydrophilicity of hydrogels is the primary factor that enables biocompatibility, thus making them a useful conduit for the fabrication of tissue constructs [55]. Hydrogels provide a suitable environment for cell growth and are highly customizable, allowing for a number of biochemical and biophysical properties to control cell functions, including cell adhesion, migration, proliferation, and differentiation [56]. Many different cell types are viable when encapsulated within hydrogels. These cell types include, but are not limited to: fibroblasts, chondrocytes, hepatocytes, smooth muscle cells, adipocytes, and stem cells [54].

Hydrogels are typically composed of either natural or synthetic polymers [19]. Natural and synthetic materials are used to produce hydrogels with various features and behaviors [50]. Recently, synthetic hydrogels have become more readily adopted than natural polymers because of their greater water absorption capacity, longer shelf life, and wide varieties of chemical resources that are available [19]. Section 2.3 and Section 2.4 describe natural and synthetic hydrogels, respectively, and delve into the details of some examples of each type. In addition, combinations of hydrogels and modifications to them will be described. Modifications allow for variations in the chemical functionalities and mechanical properties of hydrogels [57]. In synthetic hydrogels, modifications are crucial to improve their biocompatibility and cellular adhesion properties, while in natural hydrogels, modifications provide increased design capabilities. Chemical modifications of hydrogels can play a role in forming stable hydrogel networks by improving properties, like dynamic bonding, shear-thinning, self-healing, and supporting both ionic and covalent cross-linking [19].

### 2.3. Natural Polymer Hydrogels

Natural polymers are bio-derived from natural sources. These biopolymers include polysaccharides, glycosaminoglycans, and polypeptides [19]. Commonly used natural polymers include collagen, gelatin, alginate, and fibrin [19]. Natural polymer hydrogels have better biological properties than their synthetic counterparts, including increased biocompatibility, biodegradability, and cell affinity [50]. This is due to the fact that natural polymers can cover the surface of eukaryotic cells and interact with proteins to form a natural extracellular matrix [19]. Additionally, most natural polymers contain bioactive moieties that play a role in amplifying extracellular signaling to promote cell proliferation, differentiation, and function [19]. These moieties include protein ligands and cell binding motifs.

#### 2.3.1. Agarose

Agarose is a natural polysaccharide sourced from seaweed. It is not used as extensively for bioprinting applications as other naturally sourced hydrogels, as it is difficult to render printable and, being sourced from a plant, is not biomimetic for mammalian cell types [26]. However, its favorable gelation characteristics make it an attractive hydrogel component and support structure. In jet-based bioprinting, agarose was first used by Xu et al. in 2005, who printed Chinese hamster ovary (CHO) cells into an agarose substrate [23]. More recently, Gong et al. (2021) used agarose in a composite hydrogel, otherwise comprised of gelatin and alginate with adipose-derived stem cells (ASCs) suspended therein [27]. This group was able to print highly accurate and stable bioprinted structures and found that the addition of agarose increased the pore size and number in the hydrogel, which is conducive to cell proliferation [27]. Other groups have successfully used agarose in a more indirect manner. Mirdamadi et al. (2019) have described a method of embedded bioprinting, based on the pioneering work by Hinton et al. [58], whereby a cell ink is printed into an agarose slurry [59]. The agarose slurry provided temperature-resistant structural support to the soft bioprinted structures during and after printing, allowing the printed structure to remain in the slurry even when placed in the incubator. In addition, agarose gel is permeable to components of cell medium, leading to medium exchange via diffusion near the printed construct without disturbing the structure [59]. Duarte Campos et al. (2016) describe using agarose in conjunction with collagen in a jet-based bioprinter [24]. A weblike matrix is formed when agarose is added to collagen. Favorably, agarose does not change the structural topography of the collagen network, and collagen does not disrupt the gelation of the agarose. The addition of agarose to the cell ink allowed for greater viscosity, smaller drop size, and improved printer accuracy [24]. In a very recent extrusion-based bioprinting application, Dravid et al. (2022) characterize agarose-gelatin hydrogel blends and describe desirable mechanical and rheological properties for bioprinting [25]. In addition, this group was able to print SH-SY5Y cells, which differentiated into neuronal-like cells using the described agarose-gelatin cell ink.

#### 2.3.2. Collagen

Collagen type I is one of the most abundant fibrous proteins in the extracellular matrix and is the primary structural element of the extracellular matrix, providing tensile strength, regulating cell adhesion, and supporting cell proliferation [32,60]. These attributes make collagen an attractive hydrogel for use in cell inks during bioprinting. The main sources of type 1 collagen include porcine skin, rat tail tendon, and bovine skin [31]. However, collagen is limited in its use in cell inks due to its long gelation time, which can take up to half an hour at 37 °C [32]. This long gelation time can cause nonhomogeneous cell distribution as well as loss of structural fidelity in the finished print [32]. In addition, collagen is liquid at low temperatures and forms a fibrous structure with increased temperature or neutral pH, potentially proving troublesome during jet-based printing, which sometimes utilizes heat to create the jetting mechanism [34]. Lee et al. (2019) describe using freeform reversible embedding of suspended hydrogels (FRESH) to bioengineer components of the human heart at various scales [61]. FRESH works by extruding a collagen-based cell ink within a thermoreversible support bath composed of a gelatin microparticle slurry that provides support during printing and is subsequently removed. Figure 1 shows a left ventricle of the heart using human stem cell–derived cardiomyocytes, printed using the FRESH method. A dual-material printed method, seen in Figure 1A, is used to deposit the collagen ink and a cell ink with a high population density to create the ventricle. The other subsections of this image show the wave propagation behavior that could be observed in the completed ventricle, including synchronized contractions, directional action potential propagation, and wall thickening behavior typical of a ventricle [61]. Gibney et al. (2021) describe an aerosol jet bioprinting method (AJP) for the printing of dense collagenous tissues [62]. AJP is a printing method that forms an aerosol from an ink and carrier gas, and forces the aerosol to coalesce on a substrate via impaction [62]. This method could provide an interesting means of printing collagen into dense constructs as a substrate for cells, though some sources report that high density collagen constructs can limit cell proliferation, and hinder the ability to differentiate and diffuse waste products [17,63]. Conversely, one study shows that fibroblasts cultivated in high-density collagen gels (40 mg/mL) reach high viability over seven days of cultivation [64], underlining the possibility of applying AJP as a novel means of producing substrates for bioprinting. Thus, it is clear that further trials are needed.

#### 2.3.3. Fibrin

Fibrin is a fibrillar protein formed from fibrinogen (the pre-polymer form of fibrin) circulating in blood, often sourced from the plasma of mammals. In fact, a fibrin clot is the body’s first reaction to a wound, as it forms a matrix of fibers in an effort to stem bleeding. Fibrin as it is used in tissue-engineering is produced in the same manner the body forms it—fibrinogen monomers are activated to form a polymeric fibrin matrix [26]. Fibrin is broken down in the body by fibrinolysis (primarily via plasmin)—in vitro, cells produce enzymes that break down fibrin [65]. Thus, fibrin hydrogels lack structural stability for applications where they are in direct contact with cells. In addition, the high viscosity of fibrin makes it a challenging candidate for jet-based bioprinting, and fibrinogen supplies little structure and shape fidelity [65]. Thus, fibrin can be a troublesome material in cell inks. To overcome these limitations, a number of strategies can be employed to successfully integrate fibrin into cell inks for use in bioprinting. One method is to make use of fibrinogen, which has a viscosity similar to that of water. After depositing the fibrinogen, the crosslinking agent (thrombin) can be added upon (or as a substrate) the fibrinogen create a final fibrin structure via crosslinking of the fibrinogen via a calcium-dependent pathway [37,66]. This method allows for use with jet-based bioprinters, as shown by Cui et al. (2009), who printed human microvascular endothelial cells (HMVEC) with fibrin to bioprint microvasculature using this method [66]. De Melo et al. (2020) describe using an extrusion-based means of printing fibrinogen-based cell ink into a PEGDMA-alginate bath supplemented with thrombin, as to crosslink the fibrin [65]. Using this method, this group was able to bioprint a soft microenvironment that simulates the soft pericellular matrix of cartilage. This allows better nutrient transport in a bioprinted cartilage structure and thereby enables the creation of cartilage that closely simulates natural cartilage. Schöneberg et al. (2018) describe a jet-based method of bioprinting the tri-layered vessel wall to create a vessel model [37]. After printing a sacrificial gelatin core laden with human umbilical vein endothelial cells (HUVEC), a fibrin-based ink was bioprinted onto the gelatin core (lumen). Then, the gelatin was dissolved and the endothelial cells once suspended in it were allowed to sediment and attach along the lumen of the fibrin-based vessel construct [37]. This way, this group was able to make use of the glue-like properties of fibrin, as well as the stability and ideal environment for cells it provides. Despite requiring more complex printing methods than other hydrogels, benefits of fibrin include biodegradability, adhesive properties, tunable mechanical and nanofibrous structural properties [65].

#### 2.3.4. Gelatin

Gelatin is another common component of hydrogels. Because of its poor rheological properties, gelatin is often chemically modified or used in combination with another polymer before its formed as a hydrogel [50]. In several studies, gelatin is modified with furfuryl groups to make furfuryl-gelatin (f-gelatin) [50]. Furfuryl-gelatin is able to be rapidly cross-linked in the presence of visible light, maintaining its structural fidelity following cross-linking [67]. Furfuryl-gelatin can also be modified with hyaluronic acid to provide better viscosity and shear thinning, as well improve the structural integrity and stiffness of the cross-linked structure [67]. The modification of gelatin with radically cross-linkable methacryl groups, therein creating gelatin methacryloyl (GM or GEL-MA), is one approach of stabilizing gelatin, allowing it to be used in cell inks for bioprinting, among other tissue engineering applications [68]. Park et al. (2020) describe rheological methods to study the effects of cooling and heating rates on sol–gel and gel–sol transitions in GEL-MA [69]. These authors were able to confirm that cross-linking chemically modified gelatins at low temperatures can yield higher modulus (strength) than that cross-linked at a high temperature [69]. Hence, the properties of the final hydrogel are heavily dependent on the temperature at which they are processed and can be tuned to the desired application. Sharifi et al. (2021) describe the chemical modification of a gelatin-based hydrogel with glycidyl methacrylate, creating an elastic protein-based hydrogel (GELGYM) that this group used specifically for ocular tissue engineering applications, but can potentially be used for other tissue types [70]. This group also engineered an artificial blood vessel and showed that it can tolerate pressures as high as 350 mmHg before failure, making GELGYM an interesting candidate as a cell ink for bioprinting vasculature [70]. Leucht et al. (2020) describe using a combination of methacryl-modified gelatin (GM), non-modified gelatin, and acetylated GM to print vascularized bone structures [39].

#### 2.3.5. Alginate

Alginate is a natural linear polysaccharide copolymer which is extracted from brown seaweed algae and is one of the most readily used natural polymers for bioprinting [50]. It possesses several advantages as a cell ink, including being non-immunogenic, biodegradable, and non-cytotoxic, as well as being low-cost and having a rapid gelation characteristic [71]. However, it has the downsides of low cell adhesion and poor support for cell proliferation [72]. In addition, it suffers from poor printability and though it is biodegradable, it can present complex biodegradation mechanics. Fortunately, alginate is hydrophilic, and can thus be mixed easily with a number of natural and synthetic polymer cell inks to provide a more suitable habitat for cells than alginate alone can provide. Examples include, but are not limited to, silk fibroin [73], decellularized and solubilized extracellular matrix (dECM) [74], and collagen [75]. The combination of these materials allows for the balance of biological and physical properties, whereby alginate often plays a role as a structural stabilizer and thickening agent. One notable chemical modification seen in literature to improve the properties of alginate is oxidizing alginate. Oxidized alginate (ox-alg) has a faster degradation rate and more reactive groups, making alginate better suited to maintain cell function [50]. Another common alginate modification is methacrylated alginate (MeAlg/AlgMA) [50]. Methacrylated alginate has the ability to photocrosslink, which allows for greater design capabilities in adjusting the hydrogel’s mechanical properties, pore size distribution, and degradation rate [76]. In addition to modifying the hydrogels themselves, new methods are being tried to improve printing results. Hong et al. (2015) used a combination of PEG and alginate to create a highly durable and stretchable hydrogel [29]. Printed structures are rendered exceptionally robust through the addition of a nanoclay.

Recently, Teo et al. (2020) have created microstructured alginate hydrogels via a micro-reactive inkjet printing technique, whereby a precursor and crosslinker collide midair in the printing process [77]. This novel technique provides a novel means of jet-based bioprinting and shows good characteristics in the deposited alginate [78]. Usually, alginate is bioprinted using one of two methods: printing the alginate into a bath containing the crosslinker (usually calcium) or printing the crosslinker on top of the deposited alginate [77]. This method is described in detail in Section 7. The resulting printed hydrogels created using the method presented by Teo et al. can be seen in Figure 2 below. Figure 2e,f are particularly exciting prospects, as they show the possibility of bioprinting free-standing, small diameter vasculature.

#### 2.3.6. Hyaluronic Acid

Hyaluronic acid (HA) is a linear polysaccharide, naturally found in the extracellular matrix (ECM) of cartilage and synovial fluid [79]. It protects the joint by increasing the viscosity of the synovial fluid and rendering cartilage more flexible. Thus, HA is very biocompatible and supports cell signaling, wound repair, and matrix organization [80]. Additionally, hyaluronic acid has been shown to have anti-inflammatory effects, making it an interesting material when considering implantation of bioprinted structures [81,82]. Due to its negative charge, HA attracts cations and draws water through osmosis, creating a gel [26]. However, HA is highly soluble at room-temperature, limiting its structural fidelity and strength. HA can be chemically modified with a number of functional groups to limit degradation and increase stability [80]. Our group has had success (unpublished) with using a jet-based bioprinter to print a thiol-modified hyaluronic acid and thiol-modified collagen hydrogel, though extensive dilution and cooling is necessary to jet this hydrogel substrate without clogging the printer nozzles. However, despite dilution, this hydrogel crosslinked easily at room temperature and provided a supportive substrate for subsequent jetting of cells. Janarthanan et al. (2022) describe alginate-hyaluronic acid hydrogels crosslinked through several methods, namely via acyl-hydrazone, hydrazide interactions and calcium ions [40]. This group produced an “A5H5 (Alginate-acyl-hydrazide:HA-monoaldehyde, ratio 50:50) gel”, which showed a gelation time of ~60 s, viscosity of ~400 Pa s (at zero shear rate), high stability in various pH solutions and increased degradation time (>50 days) [40]. In addition, complex structures such as small, hollow tubes could be printed with ease. Noh et al. (2019) report bioprinting lattice bone structures using a cell ink comprised of HA, hydroxyethyl acrylate (HEA) and gelatin-methacryloyl [83]. This group reported stable rheological properties and excellent biocompatibility.

#### 2.3.7. Matrigel™

Matrigel™ is the trade name for the basement membrane matrix derived from the Engelbreth-Holm-Swarm (EHS) mouse tumor (sarcoma). It is a combination of proteins and small molecules, primarily collagen IV, perlacan, laminin and growth factors, and resembles the extracellular environment found in many tissues [26]. Matrigel™ is most often kept at 4 °C (liquid) and then polymerizes at body temperature—at 37 °C [26]. This characteristic makes this hydrogel an excellent contender for bioprinting applications. It is commonly used in cell culture applications as it strongly promotes cell proliferation and differentiation. In fact, cells cultured on Matrigel™ exhibit complex cellular behaviors that are otherwise difficult to induce in lab settings. For example, endothelial cells create vessel structures on Matrigel™ substrates, but not on plastic surfaces [84]. In addition, Matrigel™ is used to screen drug molecules and to observe migratory behavior of cells, such as in tumor cell metastasis. Reyes-Furrer et al. (2019) describe bioprinting pure Matrigel™ suspensions containing human skeletal muscle precursor cells [42]. This group used a cooled printhead to allow gelling of the hydrogel only upon deposition onto the printing platform at room temperature. Skeletal muscle tissue developed after culturing the printed constructs, exhibiting contractile, striated myofibers that contracted upon electrical pulse stimulation. This sort of bioprinted microphysiological system (MPS) is useful in drug development—for example, to test drugs against muscle wasting disorders [26]. De Stefano et al. (2021) used a modified extrusion bioprinter to bioprint murine prostate cancer cells suspended in Matrigel™ [41]. This group used a volumetric dispensing system to minimize erratic “splattering” extrusion that can occur when printing pure Matrigel™. While Matrigel™ does have useful characteristics with regard to cell proliferation, it requires some modification with regard to printability.

### 2.4. Synthetic Polymer Hydrogels

For tissue engineering, it is ideal to simulate the extracellular matrix (ECM) to create an ideal tissue substitute. Though synthetic hydrogels have the benefits of being capable of photopolymerization and have highly adjustable mechanical properties, they still cannot simulate the ECM as they are bio-inert [46]. Simulation of the ECM is necessary, as the ECM is not only a structural scaffold, but also a modulator of cellular behaviors, such as cell migration, proliferation, and differentiation [85]. ECM-mimetic modification of synthetic hydrogels has emerged as an important method to induce the desired cellular responses. Synthetic polymers are human-made under artificial conditions [50]. Plastics, elastomers, and synthetic fibers are most commonly employed as skeletons to develop synthetic hydrogels. Synthetic hydrogels can easily be produced and chemically modified, allowing them to be useful in specific applications. From 2009 to 2014, natural polymer hydrogels have been discussed more in the literature due to their favorable biological properties [50]. Additionally, since 2016, synthetic polymer hydrogels have experienced an upward trend in favor of natural polymer hydrogels [50]. A possible reason for the recent popularity of synthetic polymer hydrogels is the easy industrial production, as well as their capability of being highly modified, thus providing more geometries for creating tissues.

#### 2.4.1. Poly(ethylene Glycol) (PEG)

Poly(ethylene glycol) (PEG) is a very versatile synthetic compound that is favored in biomedical research because of its easily modifiable nature. In its simplest form, PEG is a polymer of ethylene oxide monomers [26]. However, through different levels of polymerization and different molecular weights, the mechanical characteristics of PEG can be changed significantly. Additionally, the polymer can take a variety of names based on the molecular weight: PEG (Mw < 20 kDa), poly(ethylene oxide) (PEO) (Mw > 20 kDa) or poly(oxyethylene) (any Mw) [26]. PEG is not viscous as a precursor solution, making it an attractive base material for cell inks, as it can be modulated specifically for tissue engineering applications. Photopolymerization is the most common method of preparing PEG hydrogels, utilizing light to convert liquid PEG macromer solutions into solid hydrogels [46]. PEG acrylates are often used in photopolymerization, including PEG diacrylate (PEGDA), PEG dimethacrylate (PEGDMA), and multi-arm PEG (n-PEG) acrylate (n-PEG-Acr) [46]. Skardal et al. (2010) describe using tetrahedral PEG tetracrylates (TetraPACs) in an extrusion-based bioprinter [44]. This group was able to bioprint hollow vascular conduits using thiolated hyaluronic acid crosslinked with TetraPAC, a PEG derivative, along with agarose microfilaments. Murine fibroblast (NIH 3T3) cells were encapsulated in this hydrogel mixture and showed good viability [44]. Gao et al. (2015) used peptide-conjugated PEG to print human mesenchymal stem cells (hMSCs) [45]. The resulting prints demonstrated excellent biocompatibility and little clogging was seen in the jet-based bioprinter used [45]. Adding peptides to PEG has proven to improve cell adhesion and support various immunomodulatory effects elsewhere [86].

Cui et al. describe using a cell ink comprised of poly(ethylene glycol) dimethacrylate (PEGDA) hydrogel and human chondrocytes for cartilage regeneration in a jet-based bio-printing process, allowing for simultaneous photopolymerization and printing [87]. This work makes use of the fact that PEG hydrogel is biocompatible, can be cleared from the body, and does not appear to alter the chondrocyte phenotype [87]. Most importantly, the compressive modulus of PEG hydrogel is tunable to match that of human cartilage, as shown by Bryant et al. [88]. Recent research concerning the tunability of hydrogels, specifically biodegradable PEG-based synthetic hydrogels, is presented by Xu et al. [89]. This group used a polycaprolactone–poly(ethylene glycol)–polycaprolactone (PCL–PEG–PCL) mixture to form a hydrogel with high elasticity and flexibility, allowing for bioprinting with a visible-light curing mechanism to print mouse fibroblasts (3T3) with an extrusion-based printer [89]. However, a low rate of degradation in vivo of PEG has been reported [90,91]. Thus, tuning PEG to modify it for use in bioprinting is very much a current topic and one that has already yielded promising results.

#### 2.4.2. Pluronic^®^

Poloxamers, though most commonly known by their trade names Pluronic^®^ and Lutrol^®^, are a class of amphiphilic triblock copolymers—i.e., they are polymers with both hydrophilic and hydrophobic regions. Pluronic is thermosensitive, and the range of its sol-gel transition temperature is broad (10–40 °C) [43]. Thus, Pluronic is stable at both room temperature and at human body temperature [43]. As it is a synthetic hydrogel, Pluronic has many of the biological disadvantages that PEG hydrogels have, namely low cell adhesion and the inability to be enzymatically degraded. However, one of the significant advantages of Pluronic is that it has good shape fidelity and is therefore accurate. It offers structural support, making it a good sacrificial material as well. However, it has the propensity to dissolve in liquids, thus often making it unsuitable for long-term contact with cells. Müller et al. (2015) describe nanostructuring Pluronic in an effort to maintain the structural properties, but also to allow long-term cell culture after bioprinting [92]. This group used a mixture of Pluronic-dimethacrylate and unmodified Pluronic to create stable gels via UV crosslinking. Then, the unmodified Pluronic is eluted from the crosslinked network, so the amount of Pluronic interacting with the cells may be reduced to increase viability. Then, to compensate for the material lost during elution, methacrylated hyaluronic acid (HAMA) was added, which has the benefit of adding biological cues to the material. This group describes excellent cell viability for a Pluronic-based hydrogel. Suntornnond et al. (2017) present a highly printable, biocompatible hydrogel for printing of perfusable vascular structures comprised of Pluronic and GelMA [43]. This group observed that the more Pluronic in the cell ink, the better the printability. Pure Pluronic was used as a support material to create vascular structures. Complex patent vascular structures were achieved using an extrusion-bioprinter, and cell attachment and proliferation (HUVECs) could be observed [43].

### 2.5. Hybrid Hydrogels

Hybrid hydrogel systems are composed of more than one type of polymeric chain or hydrogel networks that are covalently crosslinked with each other and can contain both natural and synthetic polymers [53]. Hybrid bioprinting techniques are often used to create more complex constructs and allow for more design flexibility [53,93].

An example of a hybrid hydrogel includes the blending of poly(ethylene glycol) diacrylate (PEGDA) with alginate [53]. While the PEGDA networks are chemically crosslinked, the alginate polymers are crosslinked through ionotropic gelation [53]. Although these involve two different gelation mechanisms, they assemble together in a single construct that has improved fracture strength and is better able to withstand mechanical stress [53]. Another common example of a hybrid hydrogel system is the polyvinyl alcohol (PVA)/sodium alginate (SA) hydrogel [19]. This PVA/alginate mixture displayed improved viscosity and allowed for direct 3D printing of stable scaffolds through a core nozzle tip [19].

## 3. Hydrogel-Based Scaffolds

Traditional methods for fabrication of tissue engineered vascular grafts (TEVGs) rely on a scaffolding system or some form of structural support [1,18]. Endothelial cells (ECs), smooth muscle cells (SMCs) and fibroblast cells (FCs) are seeded onto a scaffold, allowing for cellular growth and adhesion to the scaffold. Previous techniques for creating these 3D scaffolds includes synthetic production via electrospinning, freeze drying, foaming, or rapid prototyping techniques and using “bio-based” decellularized donor tissue have been described in detail [94].

In 1997, Niklason and Langer succeeded in seeding smooth muscle and endothelial cells onto tubular polyglycolic acid (PGA) mesh scaffolds [95]. To increase the strength of the bioengineered vessel, the seeded scaffold was cultured with pulsatile intra-luminal pressure and flow to mimic physiological conditions. However, the biomechanical properties of the resulting construct still lacked the strength to withstand arterial pressures [95]. Since then, the construction of vascular grafts has continued to evolve: the generation of biosynthetic vascular grafts using a 3D matrix as a scaffold is a common practice [1]. Scaffolds are biodegradable and degrade as the cells grow and form the TEVG [1]. This process may take 8–10 weeks or more [96]. Figure 3 shows the bioprinting and conventional tissue engineering method using a scaffold to create a vascular conduit or blood vessel.

However, the use of bioprinted structures or scaffolds using hydrogels presents a number of limitations in many clinical applications. First and foremost, as a scaffold degrades and is reabsorbed by the body following implantation, it often results in inflammation and disturbs tissue regeneration [1]. Unwanted degradation byproducts can be harmful to the surrounding tissue [18]. The repeated mechanical failure such as an inability to withstand high pressures and the potential for rejection of scaffold-based grafts has motivated research on scaffold-free bioprinting [18]. Producing a structurally sound tissue construct without a preexisting 3D support requires the use of cell inks with an array of mechanical and material properties. The use of cell-laden hydrogel cell inks has shown promise in the fabrication of grafts through direct extrusion methods [97]. Most commonly, endothelial and smooth muscle cells are incorporated with a hydrogel. The hydrogel itself has the task of providing a support structure for a print. To accomplish this, hydrogels often contain alginate, fibrin, PEG, and gelatin as well as post-processing crosslinking agents.

For clinical applications, using the described scaffold method has proven to still take too long for most patients due the amount of time necessary to culture patient-derived cells and form a viable vessel [96]. However, in 2011, Quint et al. found that acellular grafts can be produced by first decellularizing a bioengineered vessel and then seeding the remaining structure with endothelial progenitor cells (EPC) or endothelial cells (EC) derived from the patient [96]. This technique greatly reduces the wait time for patients and the vessels have proven successful in avoiding clotting and intimal hyperplasia.

Instead of seeding cells onto a scaffold, as previously shown, other groups have found success with using novel bioprinting technology to adhere cells to similar 3D structures [1]. A 2007 study conducted by De Coppi et al. found that 3D printing human amniotic fluid stem cells onto a scaffold resulted in 3D constructs that eventually became dense bone tissue [98]. These results now open another new direction for bioprinting that focuses on the cell type that will be deployed, rather than the substrate that the cells are being printed into.

## 4. Bioprinting Using Hydrogels

3D-printing, otherwise known as additive manufacturing, is an iterative fabrication method used for constructing three-dimensional objects through controlled material (or ink) deposition in successive layers, guided by the predefined digital 3D model, until a final three-dimensional structure is achieved [99]. Since its inception, the impact of 3D printing technology has grown rapidly in medical and scientific research and has expanded to include the printing of biological materials, namely, cells, biocompatible materials, and components that support functional living tissue [17,100,101].

The inclusion of biological materials in additive manufacturing, referred to as biofabrication, vastly complicates the 3D printing process and the nature of the printing materials. The printing process must maintain cell-compatible conditions, maintain a narrow temperature range, and not expose the cells to excessive mechanical shear forces [21]. These materials are often referred to as “bioinks”, or, as we will refer to them here, “cell inks”, as they incorporate cells with some substrate that the cells are either mixed with or printed onto during printing [4]. The cell inks must be able to act as a medium in which cells can be suspended, but also allow for printing and solidification post-print to maintain shape fidelity and mechanical support as found in the native tissue that they are replacing.

However, each bioprinting technique is limited by the properties of the cell ink. These cell ink properties include viscosity, cell seeding density, temperature sensitivity and susceptibility to shear stress, as well as the degree of thixotropy/rheopexy [102]. Table 2 displays these some of characteristics in the two most commonly used bioprinting modalities: jet-based and extrusion-based printing. A general visualization of these printing modalities as well as the bioprinting process can be seen in Figure 3. We have elected not to focus on stereolithographic bioprinting, laser-assisted bioprinting, or alternative bioprinting methods in this work in an effort to focus on the hydrogel-based cell inks and printing methods that produce large-scale prints with the potential for in vivo implantation. We refer to other works that describe these bioprinting modalities in greater detail [102,103].

## 5. Extrusion-Based Bioprinting

3D printing allows for the fabrication of 3D constructs with “high precision, repeatability, and reproducibility” [97]. The printing of living cells and biological materials has the ability to produce tissues and organs that can be used for implantation [1]. There are three primary categories of bioprinting techniques: inkjet, extrusion, and laser-assisted [1,97].

Extrusion bioprinting utilizes pneumatic pressure or a syringe pump to continuously push material through a micro-nozzle [109]. Bioprinters may include a singular nozzle or multiple extrusion sites [18]. The amount of hydrogel and/or cell ink extruded is controlled via the pressure system or position of the pump. Cell-laden hydrogels are dispensed to form 3D constructs. This can be accomplished by printing multiple layers stacked on top of one another or by extruding onto a pre-existing structure such as a scaffold. To obtain a cylindrical construct, cell inks are extruded using a coaxial printing needle or deposited directly onto a rotating rod or mandrel [18].

A significant advantage of using extrusion-based bioprinting is the ability to select a greater range of biomaterials [109]. For extrusion-based bioprinting, it is necessary for the cell ink to be stabilized upon deposition so that the extruded cell ink may hold its shape. This is often achieved through the use of high viscosity cell inks or through the use of low viscosity cell inks that are cured upon deposition by an external mechanism [102,110]. Based on the size of the micro-nozzle and the mechanical force generated by pressure or a pump, bio inks with higher viscosities are able to be printed. Extrusion-based bioprinting has a much larger working viscosity range than other bioprinting techniques, with cell inks having dynamic viscosities, and successfully being printed with a spatial resolution of ≈200–2000 µm [102]. Due to the range of possible viscosities, a number of hydrogels are compatible with extrusion-based bioprinting. Further details with regard to the characteristics of extrusion-based bioprinting can be found in Table 2.

However, extrusion bioprinting has been found to result in decreased cell viability [1]. The forceful extrusion of high-viscosity cell inks can result in cell membrane damage, depending on the extrusion pressure and micronozzle diameter. Increasing the dispensing pressure corresponds with higher rates of cell death, while increasing nozzle diameter results in increasing cell viability [106]. To fabricate complex tissues, a more intricate cell ink is necessary during fabrication. Complex cell inks typically correspond with multiple nozzles, increasing the overall difficulty of the print [4]. Additionally, the use of hydrogels in extrusion bioprinting poses potential insufficiencies in the structural support available. Due to the liquid components innate to hydrogels, the fabrication of a 3D construct with a hydrogel may lack biomechanical properties. However, the addition of crosslinking methods has been demonstrated to improve the strength of the construct [18].

In order to control extrusion and combat damage to cells during extrusion-based bio-printing, Ouyang et al. (2017) describe extruding low-viscosity methacrylated hyaluronic acid (HA) and GelMA cell inks using a transparent printer nozzle that enables photocrosslinking (visible and UV) during extrusion, immediately before deposition, allowing for the printing of structurally stable filaments [111]. This way, this group was able to print patent vascular structures onto a metal rotating rod without the limitations associated with cell ink viscosity.

Mandrels have been used previously to create tubular structures for the formation of TEVGs. One such use is for tissue engineering by cell self-assembly [1]. In a study conducted by L’Heureux et al., the pioneers of this method, a Teflon-coated stainless steel support tube was used to create the shape of a human engineered blood vessel [112]. By wrapping a sheet of fibroblasts (FCs) around the tube for three revolutions, the conduit was able to mature while maintaining the desired structure. However, self-assembling vessels require 6 to 9 months of in vitro culture and cost > $15,000 per graft, thus limiting their use [7,99].

Extrusion bioprinting was used by our group to form scaffold-free cylindrical vessels. Using an Organovo dual-head printer, we extruded a cell ink with smooth muscle cells (SMCs) and FCs onto a rotating mandrel [4]. This methodology allowed for a faster fabrication time and generated a greater number of viable prints. Without the degradation of a scaffold post implantation, these vascular grafts are not limited by the same inflammation and risk of infection [1].

## 6. Scaffold-Free Bioprinting

A 2012 study by Marga et al. demonstrated the use of hydrogels in scaffold-free grafts. The group used a dual-head bioprinter to simultaneously extruded a cell-inert hydrogel called NovoGel with a cell ink containing vascular smooth muscle cells (SMCs), endothelial cells (ECs), and dermal fibroblasts (FCs) [113]. The vascular constructs were formed by extruding onto a rod followed by maturation in a bioreactor. The use of a rod allows for the adjustment of the vessel’s diameter. In this case, the hydrogel served as a temporary support structure that was removed once the post-printed structure was fully formed, leaving behind a graft containing solely cells and cell-produced ECM [113].

A similar method for using hydrogels was performed in 2015 by Kucukgul et al. [114]. Once again, a temporary hydrogel support structure was combined with a cell ink. This study used mouse embryonic fibroblast (MEF) cells in their cell ink and NovoGel for the hydrogel component. The print was generated in a layer-by-layer fashion using Computer Aided Design (CAD) software, similarly to conventional 3D printing, as opposed to the rod used by Marga et al. The hydrogel provided a surrounding support for the formation of biomimetic aortic vascular constructs [114].

## 7. Jet-Based Bioprinting

Jet-based bioprinting, sometimes referred to as drop-on-demand (DOD) printing, was first described using modified office inkjet printers and cartridges in 2003 [15,23,115]. An inkjet bioprinter deposits small droplets of cell ink—outputting a volume of between 1–100 picoliters with a droplet diameter in the range of 10–50 μm—in a predefined geometry on a substrate or a dish [116]. Thermal and piezoelectric approaches are most commonly used to implement a jet-based bioprinting mechanism [34].

Piezoelectrically driven inkjet bioprinters use piezoelectric crystals in the printer tip, creating acoustic waves that force the cell ink through the nozzle [116,117]. An inkjet bio-printer driven by a thermal process heats a small volume of the cell ink to the temperature range of 200–300 °C for a few microseconds to form a small bubble of vapor. The resulting pressure forces a small volume of the cell ink through the nozzle, depositing a controlled spray of cell ink directly onto a dish or into a substrate, such as a hydrogel [118]. The aforementioned high temperatures raise concerns about protein denaturation and cell stress and thus, cell viability, in addition to potential cell shearing concerns due to the small opening in the nozzle of the print head. Despite these concerns, several sources present results showing high cell viability post-print, thus implying that the cells experience less stress than presumed, due to the short period of exposure to high temperatures (approximately 2 μs), though this may be dependent on the cell type [23,115,116,119]. Xu et al. (2006) have been able to show that jet-printed cells, in this case neurons, maintain the phenotypic and electrophysiological characteristics of the printed primary neurons when printed in alternating layers of a fibrin hydrogel [120]. However, attention must be paid to the formation of heat-shock proteins (HSPs) and the state of cells following printing, which can be examined per genomic analysis, for example.

The most central problem in jet-based bioprinting is that the cell ink must be able to adopt a range of states: during printing, it must be liquid to enable subsequent jetting and post-print, it must solidify into a 3D structure that maintains the desired form and provides a habitable environment for cells [17,21]. Additionally, cell inks are often limited by their viscosity: cell inks with dynamic viscosities lower than 10 mPa∙s have been reported to be compatible inkjet printing, a smaller range than what is possible with extrusion-based bioprinting [103]. In comparison with other methods, inkjet printing has the downside of low cell densities [102]. Cell concentrations are limited due to the small orifice in the tip of the printer head. The tips of these printers have a tendency for clogging, thus limiting the types of hydrogels and cell viscosities that can be used with this printing modality. Low-viscosity inks are desirable for jet-based printers, but have the troublesome side-effect of cell-sedimentation. In addition, though the motion of the printhead itself is swift, the volume extruded is lower than what is possible using an extrusion-based bioprinter. Thus, it still can take a substantial amount of time to deposit enough cell ink to generate a structure of usable size. However, jet-based bioprinting provides numerous benefits, including numerous potential cell inks, non-contact printing, which limits contamination of the substrate, control, precision, and considerable flexibility with regard to the printed geometry [117].

Novel jet-based printing types are also being developed that build upon the fundamental jet-based bioprinting principle. Teo et al. describe microreactive inkjet printing (MRIJP), whereby hydrogels are formed by in-air collision of the precursor and crosslinker in the printing process [77]. In the fabrication of alginate hydrogels, a sodium alginate precursor and calcium ions acting as a crosslinker are brought together through different methods, seen in Figure 4a–c. Figure 4a shows direct deposition of sodium alginate into a calcium ion solution bath, whereby crosslinking occurs after deposition. When using this method, the viscosity of the cross-linker bath must often be adjusted to allow for prints with good structural fidelity. Figure 4b shows the gelation of the alginate hydrogel when the crosslinker is deposited on top of the precursor, allowing for more precision, but causes heterogeneous printing on smooth surfaces, as can be seen in the corresponding figure. The MRIJP method (Figure 4c) impedes dewetting, which is the process of retraction of a droplet, effectively being the opposite of liquid spreading. It also allows for very precise deposition of the alginate hydrogel. The homogenous printed square using MRIJP, along with a schematic of the MRIJP printing process can be seen in Figure 4c. This optimization of the established jet-based bioprinting technique allows jet-based printing to be done more easily using alginate, which can be troublesome to print due to the crosslinking mechanism. In addition, this group was able to show that using this method, it is possible to print fine structures, such as free-standing tubes with small diameters (Figure 2).

In recent experiments (unpublished), our group has found high cell viability post-print using HEK cells (WT) using a jet-based bioprinter prototype. We are currently using a hydrogel mixture containing collagen, hyaluronic acid and PEGDA, which crosslinks at room-temperature, eliminating the need for an explicit crosslinking mechanism. We have repeatedly managed to successfully print HEK cells with a jet-based printer at a concentration of 3.5 million cells/mL, which is immediately followed by a Calcein AM viability assay. In our experience, cells have always survived the printing process itself, however, jet-printing does have limitations that can be observed in cell trauma post-print. One of these limitations is the fact that jet-based bioprinting is a slow building process: only few microliters of cell-ink are extruded, and several layers are necessary for a significant population of cells to be established [118]. If too few layers are printed, the cells will either die following printing due to lacking support from an aqueous environment, such as too low of a volume, or due to a cell concentration that is too low. In addition, cells are exposed to thermal stress, requiring further research, such as examining heat shock proteins (HSPs) and cell stress indicators, is necessary post-print. We have been able to print cells and hydrogels exhibiting high vitality and good structural fidelity. Despite the drawbacks described above, jet-based printing offers numerous advantages, including low cost, high resolution, high speed, and compatibility with a variety of cell inks [17].

## 8. Additives in Cell Inks

Hydrogel combinations or hybrids to combat the instability and structural weakness of many hydrogel types have been described thus far. In an effort to further stabilize and advance hydrogels for use as cell inks in bioprinting, additives are also added to the cell ink to mechanically strengthen the printed structure or improve tissue integration upon implantation of the bioprinted tissue structure.

One major challenge in tissue engineering is the poor integration of tissue-engineered constructs or the cell death in the implanted grafts due to insufficient oxygen and nutrient supply, often due to the lack of vasculature [68,121]. To induce vascularization and thereby ensure oxygen and nutrient supply, the controlled release of pro-angiogenic growth factors, such as vascular endothelial growth factor (VEGF), is a possible approach to amend these problems [68]. Claaßen et al. (2018) describe using a gelatin methacryloyl-based hydrogel (GM) as a tunable VEGF delivery system [68]. This group determined that VEGF release and the physico-chemical properties of GM/A and gelatin methacryloyl(-acetyl)-heparin methacrylate gels can be manipulated independently from each other in a broad range, allowing for various degrees of controlled VEGF release into surrounding tissues to stimulate angiogenesis [68].

“Composite” cell inks (or bioinks, as they are referred to elsewhere), are polymeric hydrogels with incorporated bioactive inorganic fillers and encapsulated cells for the use in bioprinting [122]. These inorganic fillers, either nanoparticles or anisotropic fillers, can include graphene, graphene oxide, carbon nanotubes (CNTs), hydroxyapatite (HAp) and other calcium phosphates, bioactive glasses, silica nanoparticles, and nanoclays [122]. The goal of these cell ink additives is to improve the biological and mechanical characteristics of the hydrogel-based cell ink, most commonly in extrusion-based bioprinters. One notable such composite cell ink is nanoengineered ionic–covalent entanglement (NICE) cell ink, introduced by Chimene et al. (2018) for the fabrication of both mechanically stiff and flexible structures [123]. This novel cell ink strengthening strategy combines ionic-covalent entanglement (ICE) with nanocomposite reinforcement (nanosilicates) to enhance printability and mechanical strength of the hydrogel components, namely gelatin methacryloyl (GelMA), kappa-carrageenan (κCA), and murine 3T3 preosteoblasts, of the cell ink without compromising bioactivity [123]. More recently, this group has reported using NICE cell inks to fabricate patient-specific, implantable 3D scaffolds for repair of craniomaxillofacial bone defects and were able to observe enzymatic degradability and osteoinductivity in the NICE cell ink prints [124]. NICE cell inks can be seen in Figure 5. Gao et al. were able to stimulate osteogenesis of printed bone marrow-derived human mesenchymal stem cells (hMSCs) in poly(ethylene glycol)dimethacrylate (PEGDMA) scaffolds. hMSCs were combined with PEGDMA to create the cell ink, which was then co-printed with nanoparticles of bioactive glass (BG) and hydroxyapatite (HA) and compared [125]. Though successful first results have been achieved with regard to printing hard structures with these composite cell ink techniques, little research has been done on the use of composite cell inks for soft tissue bioprinting applications and it would be interesting to see more on bioprinting of soft tissue structures. As additives and fillers in hydrogels are a quickly expanding field encompassing numerous combinations of hydrogels and additives, we refer to comprehensive publications on the subject for further reading [122,126].

## 9. Tissue Integration and Degradation

As is the case with the body’s own extracellular matrix, hydrogels are materials that are broken down over time. The degradation of hydrogels occurs primarily via enzymes, hydrolytic reactions, or ion exchange [26]. This must be considered both in the process of creating a bioprinted structure and when implanting said bioprinted structure. For example, alginate biodegrades via ionic interactions in the body, rather than via enzymatic processes as other hydrogels, such as collagen, do. This is due to the lack of the necessary enzyme, alginase, which is not found in mammals. This type of degradation must be considered in terms of bio-compatibility because alginate polymer strands exceed the filtration size for renal clearance [35] and large displacements of calcium via the ionic breakdown of alginate may lead to transient local hypercalcemia [26]. Interestingly, in some cases, synthetic hydrogels can be useful to impede the progression of inflammation. In one study, researchers found that monocyte and macrophage populations transitioned from pro-inflammatory to pro-healing phenotypes and promoted tissue revascularization when PEG hydrogels presenting adhesive peptides and angiogenic growth factors were employed [86]. Therefore, synthetic hydrogels have great potential in preventing the degradation and supporting the incorporation of prints.

Interesting novel results are presented in in vivo bioprinting. One example is presented by Skardal et al., who printed stem cells suspended in a fibrin-collagen cell ink in situ to repair skin wounds in mice [36]. With regard to work in vivo, Xu et al. describe the first in vivo monitoring of an implanted jet-bioprinted tissue, thereby demonstrating functional tissue formation and growth following implantation [127]. This group implanted both collagen/alginate composite gel seeded with cells, and gel structures without cells. They were able to see substantially more vascularization of the cell-seeded bioprinted tissue rather than the control “tissues” sans cells. Maina et al. also present excellent results regarding the implantation of bioprinted structures into rats. Good integration and patency were observed [4]. Fernandez-Yague et al. (2022) analyzed the immune response to implanted engineered hydrogels. This group explored the progression of inflammation around PEG hydrogels in great depth using Spanning-tree Progression Analysis of Density-normalized Events (SPADE) and single-cell proteomics to identify cellular processes and behavior [86]. The focus of these analyses were in vivo vascularization of the hydrogel implants and successful wound healing—previously unreported immune behaviors were reported. As summarized here, tissue integration of bioprinted constructs is quite complex and contains numerous parameters, however, the results are promising.

## 10. Conclusions

3D printing has become increasingly popular in recent years, becoming a standard practice in manufacturing and in the medical field, so it is natural that the progression of this technology grows to include the printing of biological materials, particularly cells, to create tissue. Bioprinting, however, is more complex than headlines may lead one to believe: the sheer volume of cell ink needed to print large tissue structures, especially organs, is immense. This is one of the numerous reasons that current bioprinters operates with much smaller volumina than conventional 3D printers, allowing for the small-scale fabrication of tissues. The complex nature of tissue further complicates the bioprinting of organs and large tissue structures: various tissue types are comprised of a great number of different specialized cell types and structures and perform complex physiological functions. The kidney, for example, is a much sought-after organ for transplant, as approximately 100,000 patients in the United States have been on the kidney transplant waiting list since 2017—a number that grows steadily [128]. Though it would be highly beneficial to the great number of patients to be able bioprint a kidney for implantation, this is unfortunately not possible with current technology. It is estimated that the kidney harbors more than 70 different cell types, and more types of cells continue to be discovered [129]. In addition, it is an organ with complex functional structures.

For this reason, it is, in the current state of research, advantageous to optimize bioprinting with smaller structures that (a) require a small number of different cell types (b) structures that are simple in geometry and physiological function (c) require little to no vascularization. An example of such a structure is cartilage, a non-vascularized structure with little variation in cell types and a relatively simple construction. Though larger constructs have been printed successfully, such as the heart printed by Lee et al. [61], further optimization beyond the bioprinter must be done to culture the vast volume of cells needed to create bioengineered tissue out of a scaffold that size. In addition, it is important to continue to optimize the existing methods and materials for bioprinting. Sardelli et al. (2021) present a non-traditional crosslinking method to allow for the fine control of alginate properties during printing [130]. Not dissimilar to Teo et al. [77], who developed a modification to jet-based printing for more precise jet-based printing. It is the continuation of trying new, non-traditional methods that will continue to improve the bioprinting field and allow it to grow and change.

Current bioprinting technology allows researchers to precisely deposit cells and hydrogels for the fabrication of functional tissue. Polymeric hydrogels are a significant component of cell inks because they are highly compatible, form crosslinked polymeric networks that support cells structurally and functionally, and can be induced to solidify in a controlled manner, so that several layers (and even cell types) and be iteratively constructed. The ideal hydrogel would have excellent biomimicry and biocompatibility, as well as a degradation similar to that of the ECM, and offer functional support. Some hydrogels offer more structural support, but do not provide a biologically ideal environment for cells, or vice versa. To remedy this, a number of promising materials and methods have been described here, including manipulation of known hydrogels through chemical modulation or additives, combinations of hydrogels, and novel bioprinting methods. Finally, using hydrogel-based cell inks in bioprinting is a field that is currently experiencing a surge of growth, as no “ideal” hydrogel provides all the desired characteristics of printability and providing an ideal structural and functional environment for cells. Many novel methods, such as 4D bioprintable self-healing hydrogels with shape memory and highly tunable PEG hydrogels for implantation, propel the field forward [86,131]. However, as hydrogels are so highly alterable and can be manipulated to such a high degree, it will not be long until we can print highly functional and structurally complex structures with bioprinting on a greater scale.

## Figures and Tables

**Figure 1 pharmaceutics-14-02596-f001:**
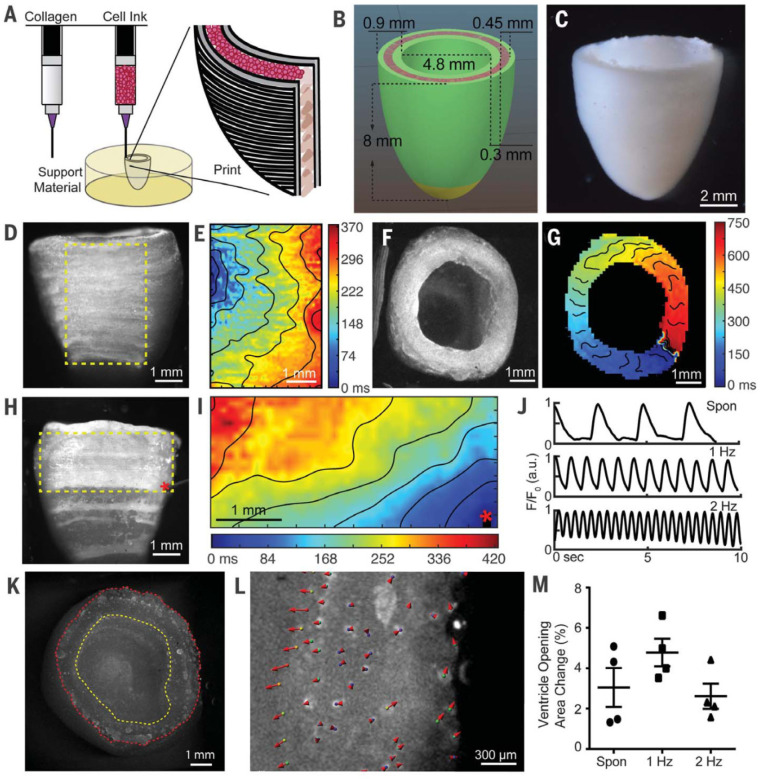
3D printing of human hearts. (**A**) Schematic of dual-material FRESH printing. (**B**) Ventricle model with cardiac cells (pink) and collagen shells (green and yellow). (**C**). Micrograph of FRESH-printed ventricle. (**D**) FRESH-printed ventricle stained with calcium-sensitive dye showing homogeneous cell distribution. (**E**) Calcium mapping showing calcium wave propagation of the boxed segment (yellow) of (**D**). (**F**) Top view of FRESH-printed ventricle stained with calcium-sensitive dye. (**G**) Calcium mapping showing circular calcium wave propagation around the ventricle. (**H**) Point stimulation of FRESH-printed ventricle stained with calcium-sensitive dye (red asterisk indicates electrode location). (**I**) Calcium mapping of the subregion (yellow) in (**H**) showing anisotropic calcium wave propagation with longitudinal conduction velocity. (**J**) Calcium transient traces during spontaneous contractions (top), 1 Hz field stimulation (middle), and 2 Hz field stimulation (bottom). (**K**) Top-down image of the FRESH-printed ventricle with inner (yellow) and outer (red) walls outlined. (**L**) Subregion of the ventricular wall analyzed for displacement during 1 Hz field stimulation, showing inner and outer wall motion upon stimulation. Magnitude and direction are indicated using arrows. (**M**) Cross-sectional area changes of the ventricle interior chamber at peak systole (*N* = 4, data means are ±SD). Reproduced with permission from [61], the American Association for the Advancement of Science 2019.

**Figure 2 pharmaceutics-14-02596-f002:**
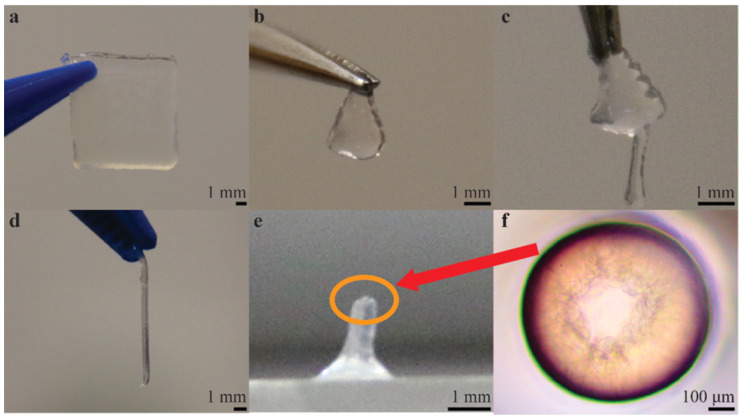
Different structures of microreactive inkjet-printed alginate hydrogels. (**a**) Alginate rectangular film (*f* = 500 Hz, 10 layers). (**b**) Alginate triangular film (*f* = 100 Hz, 20 layers). (**c**) Alginate tree (*f* = 100 Hz, 20 layers). (**d**) Alginate stick (*f* = 100 Hz, 200 layers). (**e**) Alginate hollow tube (*f* = 10 Hz, 300 layers). (**f**) Microscopic image of the hollow tube. Reproduced with permission from [77], American Chemical Society, 2020.

**Figure 3 pharmaceutics-14-02596-f003:**
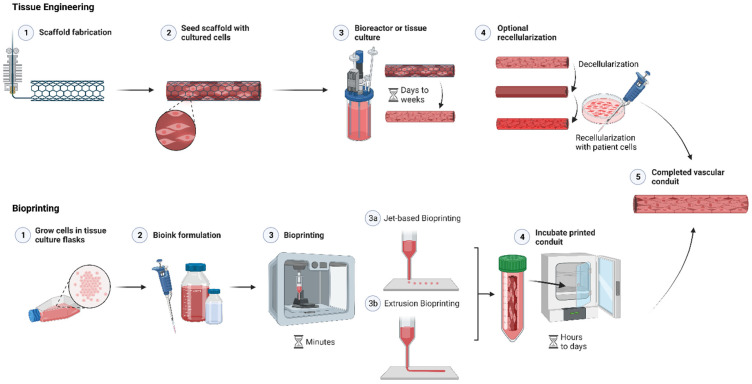
Tissue engineering and bioprinting approaches to creating vascular conduits. Created using BioRender.

**Figure 4 pharmaceutics-14-02596-f004:**
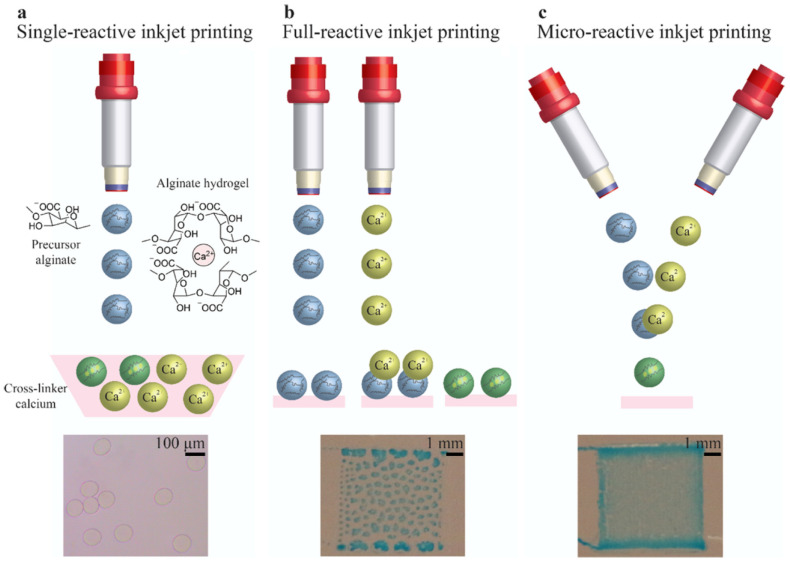
Three different strategies of jet-based bioprinting approaches to create alginate hydrogels. Blue dye is used to visualize the deposited alginate hydrogel. (**a**) Single-reactive printing involves direct deposition of the precursor into a crosslinker. (**b**) Full-reactive printing of the precursor followed by the crosslinker, creating the finished hydrogel (green). (**c**) Micro-reactive inkjet printing, whereby crosslinker and precursor meet midair, and a hydrogel is formed before deposition. Reproduced with permission from [77], the American Chemical Society, 2020.

**Figure 5 pharmaceutics-14-02596-f005:**
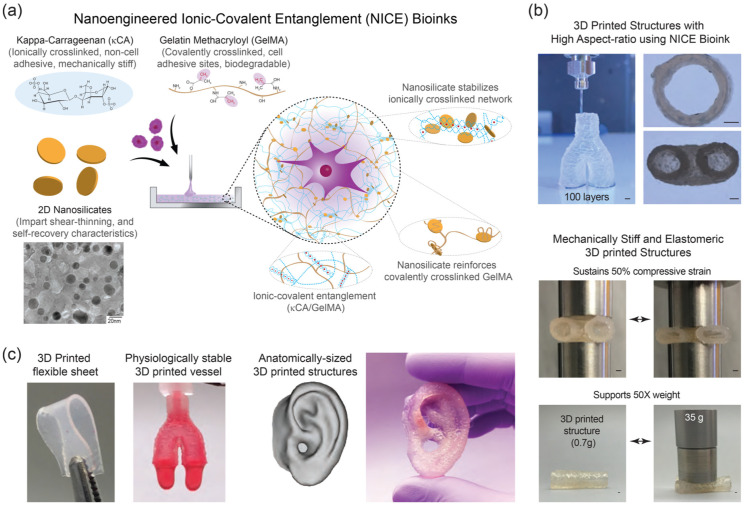
(**a**) Composition of Nanoengineered Ionic-Covalent Entanglement (NICE) cell inks, which employ nanocomposite reinforcement and ionic-covalent entanglement mechanisms to create a cell ink that is elastic, touch, and highly printable. Nanosilicates are used to reinforce an ionic-covalent entanglement hydrogel made from GelMA and κCA. These interactions allow the NICE bioink to behave as a solid at low shear stresses and improve shear thinning characteristics during bioprinting. Lower left: TEM imaging of two-dimensional nanosilicate particles show uniform morphology. (**b**) Examples of free-standing, tall structures using NICE bioink. Scale bar = 1 mm. Crosslinked structures are stiff and elastomeric, and can support more than 50-times their own weight. (**c**) Examples of structures printed using NICE cell ink that emulate anatomical structures, and are mechanically and physiologically stable and exhibit high structural fidelity. Reproduced e with permission from [123], the American Chemical Society, 2018.

**Table 1 pharmaceutics-14-02596-t001:** An assortment of polymeric hydrogels commonly used in cell inks for bioprinting.

Cell Ink	Type	Printing Modality	Biodegradation Mechanism	Advantages	Disadvantages
Agarose	Natural	- Inkjet [23,24] - Extrusion [25]	- Nonbiodegradable [26]	- Provide good structural support in hydrogel matrices [27]	- Difficult to print - Complex biodegradability
Alginate	Natural	- Inkjet [28] - Extrusion [29] - Laser [30]	- Ionic displacement [26]	- High cell viability post-print (>90%) [31] - Rapid and easy crosslinking [32] - Allows for several crosslinking types [32] - Inexpensive	- Poor mechanical strength [33] - Low cell recognition and adhesion [34] - Complex biodegradability [35]
Collagen	Natural	- Inkjet [36] - Extrusion [37] - Laser [38]	- Enzymatic (MMP) [26]	- Highly biocompatible - Easily degraded	- Poor mechanical properties - Poor cross-linking kinetics
Fibrin	Natural	- Extrusion [37]	- Enzymatic (plasmin) [26]	- Encourages cellular growth	- Liquifies at high temperatures - Crosslinking mechanism emulates nature
Gelatin	Natural	- Extrusion [39]	- Dissolution at 37 °C - Enzymatic (MMP) [26]	- Excellent biocompatibility - Simple crosslinking means	- Propensity for dissolution in body-temperature aqueous environments - Requires modification for ideal gelation characteristics
Hyaluronic acid (HA)	Natural	- Extrusion [40]	- Enzymatic (hyaluronidase) [26]	- Highly biocompatible	- Requires modification with regard to gelation
Matrigel™	Natural	- Extrusion [41] - Inkjet [42]	- Enzymatic (MMP) [26]	- Highly biocompatible - Provides an ideal environment for cell growth	- Maintains liquid form at low temperatures—may require cooling for printing - Poor structural properties during printing
Pluronic^®^	Synthetic	- Extrusion [43]	- Nonbiodegradable [26]	- Good structural fidelity	- Does not promote cell adhesion or proliferation - Biodegrades under very specific conditions
Polyethylene glycol (PEG)	Synthetic	- Extrusion [44] - Inkjet [45]	- Nonbiodegradable [26]	- Easily altered through chemical modification [46] - Hydrophilic - Addition of degradable materials is possible	- Does not provide biological cues for proliferation - Biodegrades under very specific conditions - Photocrosslinking may affect cell viability [47] - Does not naturally biodegrade

**Table 2 pharmaceutics-14-02596-t002:** Characteristics of jet-based and extrusion-based bioprinters.

Bioprinting Method Characteristics	Jet-Based	Extrusion-Based	References
Ink Viscosity	Requires lower viscosities to prevent tip clogging, ~10 mPa/s, 3.5–12 mPa∙s	Large range with higher viscosities, 30 mPa∙s to >6 × 10^7^ mPa∙s	[17,18,104,105]
Cell Viability	High, >85%	Low to Medium, 40–80%	[1,17,104,106]
Cell Seeding Density	Low cell densities to prevent clogging, <10^6^ cells/mL	Multiple cell types can be extruded (i.e., dual head extrusion) with high cell density	[4,17,104,107]
Cost	Low	Medium	[17,108]
Printing Speed	Fast	Slow	
Use	Faster operation time, low preparation time	Low to medium preparation time, low end resolution	[17,104]

## Data Availability

Not applicable.

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
