# Peer review of "3D Bioprinting Using Hydrogels: Cell Inks and Tissue Engineering Applications"

_pharmaceutics, 2022, doi:10.3390/pharmaceutics14122596_

Round 1
Reviewer 1 Report
The article "3D bioprinting using hydrogels: cell inks and biomedical applications" is composed of 24 pages, 2 Tables, 5 figures and 133 references. This review is an overview of the hydrogel use in the case of 3D bioprinting interesting for the scientific community.
This paper is totally in adequation with the journal, Pharmaceutics. Before publication, the authors need to do some modifications, especially to be more precise and state that this review focuses only on polymeric hydrogels.
Comments:
Line 50 « bio-printed » to replace “bi-oprinted”
Line 61: the authors define what a hydrogel is but only in the case of polymeric one, but hydrogels aren’t only polymeric, there is also supramolecular hydrogel based on peptides, low molecular weight gelators as bolaamphiphiles, … I think the authors need to be more exhaustive on this point to include all kind of gelators used for hydrogel formulation in general.
· It’s seems that the authors focus only on polymeric hydrogel, it must be clearly stated that this review focus only on polymeric hydrogels
Line 165 : “improve” to replace improving” isn’t it?
· Line 186: “its” to replace it’s” isn’t it?
· Line 194: “described” to replace “describe”
· When the authors present results from publications they used the present tense whereas past tense is more appropriate since the study is finished
· Line 304-305: there is one) missing after GM I think
· Line 332: The authors present the figure 3 before the figure 2, it’s quite confusing. The authors may need to make some adjustments. What is the pertinence of a sentence only saying "Figure 3 shows the printing technique”?
· Line 332/333: Two “Novel” in the same sentence, a little bit too much
· Line 433/434: “bio-printing” to replace “bi-oprinting”
· Line 471: it seems that the authors forgot to cite the figure in the text. They need to correct it
· Line 518: delete the point after 3
· Line 570: the authors need to correct the viscosity unit: Pa.s
· Line 584: “bio-printing” to replace “bi-oprinting”
· Line 596: The authors forgot to delete “note: add more on cost…”
· Line 629: “bio-printer” and not “bi-oprinter”
· Line 649: mPa.s the authors need to be careful on the unity
· Figure 4: higher image quality is required
· Line 678: why “et al.” in italic while the authors didn’t do it before in the manuscript
· Line 682: “in an effort” two times in a row. It needs to be corrected
Reviewer 2 Report
The Paper “3D Bioprinting using Hydrogels: Cell Inks and Biomedical Applications “ is a review paper about the application on two 3D printing techniques (Extrusion and Jet-based bioprinting) for the production of hydrogel-based scaffold for tissue engineering.
The introduction clearly define and limits the scope of the review, based on which the title of the paper could be slightly modified changing the general term “Biomedical application” into a more precise “Tissue engineering applications”.
After the introduction the authors organize their work starting from a general presentation of hydrogel properties (and the need of crosslinking), then they present a (indeed complete – to my opinion) survey of the different hydrogels reported in literature and then focus on 3D printing technologies, on the possibility to use composite hydrogels and on the tissue integration and degradation requirements.
Through the manuscript and in the conclusion some perspectives for the future development in the field are suggested.
Two tables are presented, which effectively sum up many of the presented concepts.
Instead, I have some concern about figures: 5 figures (4 from literature) are reported, but to my opinion they are not as effective as the tables: first of all, the captions are not self-explanatory, so understanding their meaning is really tough. Further, I suspect some mis-citation of the figures was made in the manuscript: for example in line 332 of the manuscript a reference is done to figure 3, but, out of the text in line 330 to 332 I think the appropriate figure should be figure 4. Further figure 3 (or 4) appears before figure 2, so renumbering is needed. Further, if I got it right, the figure presently indicated as 3 is never described in the manuscript.
Finally, the reference to figure 4 in line 713 is (probably) again to be corrected, as it seems that NICE cell ink prints are represented in figure 5 (by the way, figure 5 is never cited in the manuscript).
Besides this point, and the need of a careful re-read of the manuscript (for example, line 794 seems incomplete, typos are present in lines 423, 734, and an author note is present in lines 596-598), the manuscript deserves publication, maybe after having considered the comments detailed below
2.1.3 Crosslinking
On line 134 and following, the physical cross-linking is defined as “carried out by physical procedures (e.g. UV light), and involves intermolecular interactions…”. Indeed UV light is used to trigger chemical irreversible cross-linking. I suggest therefore to delete the reference to UV in the sentence.
2.3.5 Alginate crosslinking
In the last part of the section (lines 330 and following) different methods for crosslinking methods are described, both innovative and traditional. The authors may consider also other non traditional crosslinking methods which allow a fine control of crosslinking kinetics and crosslinked alginate properties, as reported in Sardelli, L. ;Tunesi, M. ;Briatico-Vangosa, F. ;Petrini, P., 3D-Reactive printing of engineered alginate inks, Soft Matter Volume 17, Issue 35, Pages 8105 - 811721 September 2021 DOI: 10.1039/d1sm00604e
5. Extrusion-Based Bioprinting
On line 568 and following the rheological requirements of extrudable cell inks are reported in terms of a range of viscosities (101-1013 Pas is indeed an interval different than what reported in table 2!), however there is some literature which details more about the requirement for hydrogels printability (for example N. Paxton et al., Proposal to assess printability of bioinks for extrusion-based bioprinting and evaluation of rheological properties governing bioprintability, Biofabrication, 2017, 9 , 044107 ).
6. Jet-Based Bioprinting
On lines 665 and following, the issue of post printing cell viability is considered, and ascribed partly to the fact that the jet-based printing process is a slow one. However, in table 2 this process is reported as “fast”, while extrusion printing process as “slow”: which of the two statements is the appropriate?
Reviewer 3 Report
The authors presented an article „3D Bioprinting using Hydrogels: Cell Inks and Biomedical Applications“. The review is interesting and deserves the attention of readers. I believe that the results are sound, and can be interesting for the Journal’s readers. The authors presented the issue of 3D printing of hydrogels, introduced the types of materials and presented bioprinting and the use of hydrogels in biomedicine.
1. Excellent Work done by the Authors. The novelty of the work is good.
2. The abstract needs not be improved.
3. The propertes, materials and using hydrogels not be improved.
4. The conclusion and Future works are good.
Review is important for inspiration and motivation for new scientific projects. I have no recommendations and the Paper is suitable for publication.
Round 2
Reviewer 2 Report
The authors revised the review based on all reviewers comments.
Still, figure 5 appears in the review, but it is not cited in the manuscript. It is clearly not necessary to the work, so I'd suggest to remove it
Author Response
Dear reviewer, we are appreciative of your comment and continued support of our work. We have cited Figure 5 in line 791 of the manuscript. We do believe that NICE inks are very relevant to the field and feel that Figure 5 visualizes this type of bioink well.